# Automated scoring of nematode nictation on a textured background

**Patrick D. McClanahan**, **Luca Golinelli**, **Tuan Anh Le**, **Liesbet Temmerman***

Animal Physiology and Neurobiology, Department of Biology, KU Leuven, Leuven, Belgium

* liesbet.temmerman@kuleuven.be

**Data Availability Statement:** All of the data collected and used in this study are available on BioStudies: https://www.ebi.ac.uk/biostudies/studies/S-BSST1149

## Abstract

Entomopathogenic nematodes, including *Steinernema* spp., play an increasingly important role as biological alternatives to chemical pesticides. The infective juveniles of these worms use nictation–a behavior in which animals stand on their tails–as a host-seeking strategy. The developmentally-equivalent dauer larvae of the free-living nematode *Caenorhabditis elegans* also nictate, but as a means of phoresy or "hitching a ride" to a new food source. Advanced genetic and experimental tools have been developed for *C. elegans*, but time-consuming manual scoring of nictation slows efforts to understand this behavior, and the textured substrates required for nictation can frustrate traditional machine vision segmentation algorithms. Here we present a Mask R-CNN-based tracker capable of segmenting *C. elegans* dauers and *S. carpocapsae* infective juveniles on a textured background suitable for nictation, and a machine learning pipeline that scores nictation behavior. We use our system to show that the nictation propensity of *C. elegans* from high-density liquid cultures largely mirrors their development into dauers, and to quantify nictation in *S. carpocapsae* infective juveniles in the presence of a potential host. This system is an improvement upon existing intensity-based tracking algorithms and human scoring which can facilitate large-scale studies of nictation and potentially other nematode behaviors.

## Introduction

Under conditions of crowding or limited food, nematodes of many species can develop into a developmental stage that is physically and behaviorally specialized for dispersal [1, 2]. Such animals are usually called infective juveniles (IJs) or dauer juveniles in entomopathogenic nematodes (EPNs) like *Steinernema carpocapsae [3],* and dauers or dauer larvae in free-living nematodes like *Caenorhabditis elegans* [2, 4].

Nictation is a dispersal behavior common to both IJs and free-living dauers. Nictating worms stand on their tails, sometimes waving back and forth, which facilitates attaching to [5], and possibly jumping onto [6] a nearby larger animal. Once attached, EPN species may infect this animal [7], and free-living species may use it as a means of transport to a new food source (phoresis) [8]. Hence, understanding nictation is of interest for both the development of biocontrol agents [9] as well as understanding the ecology and biology of dispersal [10].

**Funding:** This work was supported by the Fonds Wetenschappelijk Onderzoek – Vlaanderen (https://www.fwo.be/, FWO G085521N awarded to L.T.) and KU Leuven (https://www.kuleuven.be, C16/19/003 awarded to L.T.). The N2 strain was provided by the CGC, which is funded by NIH Office of Research Infrastructure Programs (P40 OD010440). The funders had no role in study design, data collection and analysis, decision to publish, or preparation of the manuscript.

**Competing interests:** The authors have declared that no competing interests exist.

Nictation cannot be studied on the smooth agar surfaces commonly used in nematode research because worms require a textured substrate to begin nictating [7, 11, 12]. Instead, natural or naturalistic substrates such as soil [13], sand [7], or gauze [14] have historically been used in nictation studies, but these can be difficult to replicate exactly and are not conducive to recording useful videos amenable to automated tracking analysis (*i.e.* videos in which the animals stay in the focal plane and are easily distinguishable from the background). A decade ago, the development of microdirt arenas consisting of a rectilinear array of cylindrical posts on a planar molded agar surface provided a substrate for nictation studies that is both reproducible and amenable to video recording of both *C. elegans* and *S. carpocapsae* [5, 14, 15]. Since then, a number of genes and pathways that regulate nictation have been identified, including insulin and TGF-β signaling [16], piRNAs [17], and several neuropeptides [15, 16, 18]. However, scoring nictation is still done by a human observer, often in real time. Therefore, the throughput as well as the ability to further analyze more subtle phenotypes could be improved by recording the behavior and automating the scoring process.

A major hurdle in automating the scoring of nictation is reliably tracking the worms. Videos of worms on textured substrates pose a challenge to classical worm trackers because the intensity-based segmentation algorithms (*e.g.* [19, 20]), used in all but a few published worm trackers (see [21, 22] for notable exceptions) are designed to detect worms recorded against a smooth, uncluttered background. Meanwhile, state-of-the-art object detection algorithms have improved dramatically in the past decade on benchmark tasks [23]. Recent work in the area of regions with convolutional neural network features (R-CNNs) has repopularized convolutional neuronal networks and developed them from image classification [24, 25] to object detection [26] and segmentation tools. One particularly successful deep learning model, Mask R-CNN, can detect and segment a variety of objects against a heterogeneous, cluttered background [27].

In this report, we overcome the challenges of tracking animals on a textured substrate by training a Mask R-CNN to reliably detect and segment *C. elegans* dauers and *S. carpocapsae* IJs freely behaving on a microdirt arena. We further refine these segmentations by using a simple ridgeline-based algorithm or, where necessary, a deformable model to find the centerline coordinates of each tracked worm in each frame. From these video and postural data, we compute a set of quantitative features useful for detecting nictation. Using a human-scored subset of our data, we train a neural network machine learning classifier to use these features to recognize nictation with accuracy similar to that of other human scorers. Utilizing our system, we show how the nictation behavior of *C. elegans* increases as the animals become dauers, and then remains relatively constant for weeks. We then show the effect of the presence of potential host cues on *S. carpocapsae* nictation ratio. Our pipeline, which is written in Python and can be run using a click-through GUI, can facilitate large-scale nictation studies which otherwise would require a prohibitively large amount of labor.

## Results

### Mask R-CNN can reliably detect and segment *C. elegans* on a heterogeneous background

We used recordings of *C. elegans* dauers freely behaving on a microdirt arena to develop our tracking and scoring pipeline. First, we tried using classical intensity-based thresholding to detect and segment the animals. We used background subtraction to enhance the contrast of the worms and eliminate the appearance of the microdirt posts [19], but the posts remained visible due to slight movements in the background over time (**Fig 1A, top**). We applied video

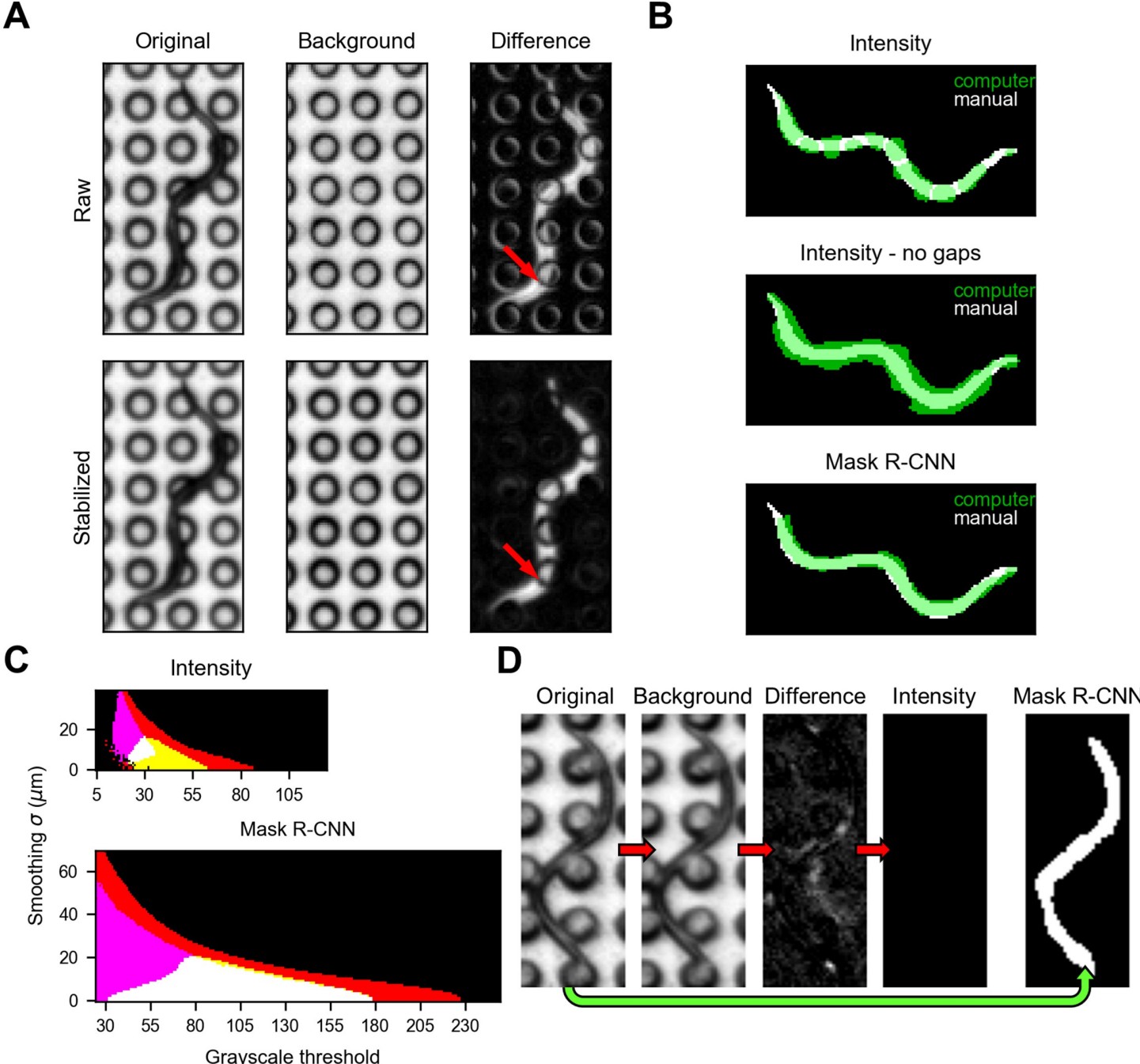

**Fig 1. Comparison of intensity-based segmentation and Mask R-CNN-based segmentation of *C. elegans* dauers on a microdirt chip.** **(A)** Effect of video stabilization on background-subtracted images. Original image (*left*), max-merge background (*center*), and background-subtracted image (*right*) without (*top*) and with (*bottom*) video stabilization. Arrow indicates one of many locations where the contrast of a post is reduced, but a gap remains in the worm, following video stabilization. **(B)** Overlay of machine vision segmentations (green) on a manual segmentation (white) of the dauer from panel A: intensity segmentation with parameters chosen to maximize IoU (*top*), intensity segmentation with parameters chosen to maximize IoU while eliminating gaps (*middle*), Mask R-CNN segmentation with parameters chosen to maximize IoU (*bottom*). **(C)** Colormap showing combinations of smoothing and grayscale threshold parameter values that result in 100% detection without gaps (red); 100% detection without gaps and $> 0.5$ IoU (yellow); 100% detection without gaps and $< 10\,°m$ centerline RMSD (magenta); or 100% detection without gaps, $> 0.5$ IoU, and $< 10\,°m$ centerline RMSD (white) of 40 manually-segmented worms. Other combinations of criteria did not occur in the parameter set tested. Results for intensity (*top*) and Mask R-CNN (*bottom*) are scaled equally. There were 106 and 1561 parameter value combinations meeting all three criteria for intensity- and Mask R-CNN-based segmentation, respectively. **(D)** Intensity-based and Mask R-CNN-based segmentation applied to a dauer that did not move during recording. In all worm images, the center to center spacing of the posts is $75\,°m$.

stabilization to the video frames to correct for these movements, but gaps remained in the animals where they overlapped the posts' dark edges (**Fig 1A, bottom**).

To segment the worms in the contrast-enhanced images, we applied Gaussian smoothing (standard deviation $\sigma$) followed by binarization at a grayscale threshold ($t$) [19, 28–30], keeping all resulting regions of interest (ROIs). To determine the best combination of $\sigma$ and $t$, we performed a grid search using the intersection over union (IoU) of the ROIs against a set of 40 manually annotated worms. We achieved a mean IoU of 0.61 (**S1A Fig**), but the gaps from the posts meant that, often, multiple ROIs pertained to one worm (**Fig 1B, top**), complicating the process of tracking and finding worm centerlines.

We tried two approaches to overcome this limitation of intensity-based segmentation. First, we eliminated gaps in the ROIs by changing $\sigma$ and $t$, optimizing for IoU but only accepting parameter value combinations that eliminated gaps. Segmentation was successful, but the resulting ROIs were enlarged, reducing the mean IoU to 0.43 (**Fig 1B, middle and S1B Fig, top**). Second, we used an alternative, deep-learning approach to segment the worms. We fine-tuned a Mask R-CNN [27] (see methods) and used it to generate grayscale images of probable worms (**S2 Fig**). We then applied smoothing and thresholding as in intensity-based segmentation. After optimizing $\sigma$ and $t$, the resulting mean IoU was 0.62, similar to intensity-based segmentation, and the ROIs did not contain gaps (**Fig 1B, bottom and S1A and S1B Fig**).

Because ROIs are the basis for obtaining descriptions of worm posture and ultimately behavior, we weighed several additional arguments to decide whether to rely on intensity or Mask R-CNN segmentation for the remainder of our work. For one, a segmented worm is used to calculate a centerline, a powerful means to describe nematode posture [31]. Even though ROI quality clearly differs between (gapless) intensity-based and Mask R-CNN methods, when $\sigma$ and $t$ were optimized for the accuracy of centerlines drawn from the resulting segmentations, the two methods performed similarly, with a slight advantage for intensity [root-mean-square deviation (RMSD) of 4.13 μm for intensity, 5.06 μm for Mask R-CNN] (**S1C Fig**).

However, there are some disadvantages of intensity-based segmentation that do not outweigh the slight centerline advantage, especially for our application. For instance, successful segmentation often requires a careful selection of parameters that have to be adjusted for minor changes in imaging conditions. To compare the stringency of choice of $\sigma$ and $t$ in the two methods, we decided on a set of three quality benchmarks that must be surpassed simultaneously for a segmentation to be deemed successful. Namely, (1) that all worms in the manually-annotated set should be detected without gaps (**S1D Fig**), (2) that the mean IoU should be at least 0.5, and (3) that the mean centerline RMSD should be less than 10˚m. Again using a grid search, we found that a 14.7x larger set of $\sigma$ and $t$ could meet these benchmarks by using a Mask R-CNN rather than intensity segmentation (**Fig 1C**). Other disadvantages of intensity-based segmentation are the need to stabilize the video, which could be difficult if displacements are larger than the inter-pillar distance, and its inability to segment worms that do not move out of their own footprint during the recording period. Mask R-CNN, by contrast, works directly on the non-background-subtracted, non-stabilized video frames and segments non-moving worms (**Fig 1D**). Considering these factors, we used Mask R-CNN for segmentation in the remainder of this study.

## Extracting postural information using ridgeline points and a deformable model

The locomotion of a nematode can be described by its centerline coordinates over time. To compute the centerline from a binary ROI, we took the distance transform (the distance of each point in the ROI to the nearest background point) [32] and found ridgeline points that

were local maxima in both the horizontal and vertical directions. We linked these ridgeline points by proximity. Unlike commonly-used skeletonization and morphological thinning algorithms (*e.g.* [29, 33]), the sets of ridgeline points rarely contain spurs or "split ends", but they also terminate before reaching the endpoints of the worm. We located the endpoints by finding minima of the interior angle of the ROI perimeter [28, 34]. We then fit a spline to the endpoints and ridge points to compute the centerline (**Fig 2A**).

This and other simple centerline-finding procedures often fail when a worm crosses or contacts itself, for example, during an omega turn (**Fig 2B**). We used such configurations, along with excessive length, excessive curvature, and self-intersection, to flag potentially incorrect centerlines (8.9% of our *C. elegans* dataset). Deformable models have proven useful for correctly ascertaining the posture of self-touching worms [35], so we designed and fit a simple deformable model to correct flagged centerlines.

Our deformable model consists of nine centerline points whose positions are initialized based on a non-flagged or corrected centerline of the same worm from an adjacent frame. A binary ROI is drawn around these points, and its width adjusted such that its area matches that of the target ROI. During fitting, these points move iteratively toward uncovered regions of the target image in a manner analogous to the pull of gravity, where the "force" on each point is the vector sum of the "gravitational pull" of each uncovered ROI pixel (**Fig 2C**). Additionally, in each iteration all points are acted upon by a net torque, rotating the deformable model as if it were a rigid body. Correcting flagged centerlines in this way improved behavioral scoring accuracy (see below).

## Feature-based scoring of nictation

We used supervised machine learning to score nictation. To create a ground truth dataset, a human (the trainer) manually scored the tracked animals in a training video (4.15 h of behavior) and a separate testing video (4.05 h of behavior). Each worm-frame was scored as nictating if greater than 1/5 of the animal was elevated off the substrate, recumbent if not, or censored if the behavior could not be determined, *e.g.* a tracking error (0.29% of the training set and 0.73% of the test set). Nictation nearly always involved elevation of the anterior portion of the worm, but occasionally the mid-section or even tail was elevated instead (**S1 Video**). Similar to previous studies utilizing feature-based behavioral classification [36–38], we computed 17 movement, postural, and other features from each worm-frame in our dataset (**Fig 3A**). For each feature, we also calculated the first derivative with respect to the preceding and succeeding frames as well as five statistical features (mean, median, min, max, and variance) in a 1.0 s (5 frame) window centered on the frame in question [39].

Using the Scikit Learn library [40], we performed five-fold cross-validation to evaluate the performance of several common machine learning algorithms in combination with several feature scaling methods. Although not the fastest, we achieved the highest accuracy using a neural network and min-max scaled features, with an average of 96.0% of worm-frames scored correctly in the validation data, and 94.9% of worm-frames scored correctly in the separate testing video (**S3 Fig**). Unsurprisingly, errors tended to occur where recumbence and nictation overlap in feature space (**S4 Fig**). We also compared the accuracy of this method on the testing video with and without using the deformable model method to correct flagged centerlines in the test dataset, using a neural network trained on the entire training video with corrected centerlines. In the test set, 4.4% of centerlines were flagged and fixed, improving accuracy from 89.0% to 95.1%.

Common metrics used to quantify nictation are nictation ratio (NR), the proportion of time spent nictating; initiation rate (IR), the rate at which non-nictating worms start nictating;

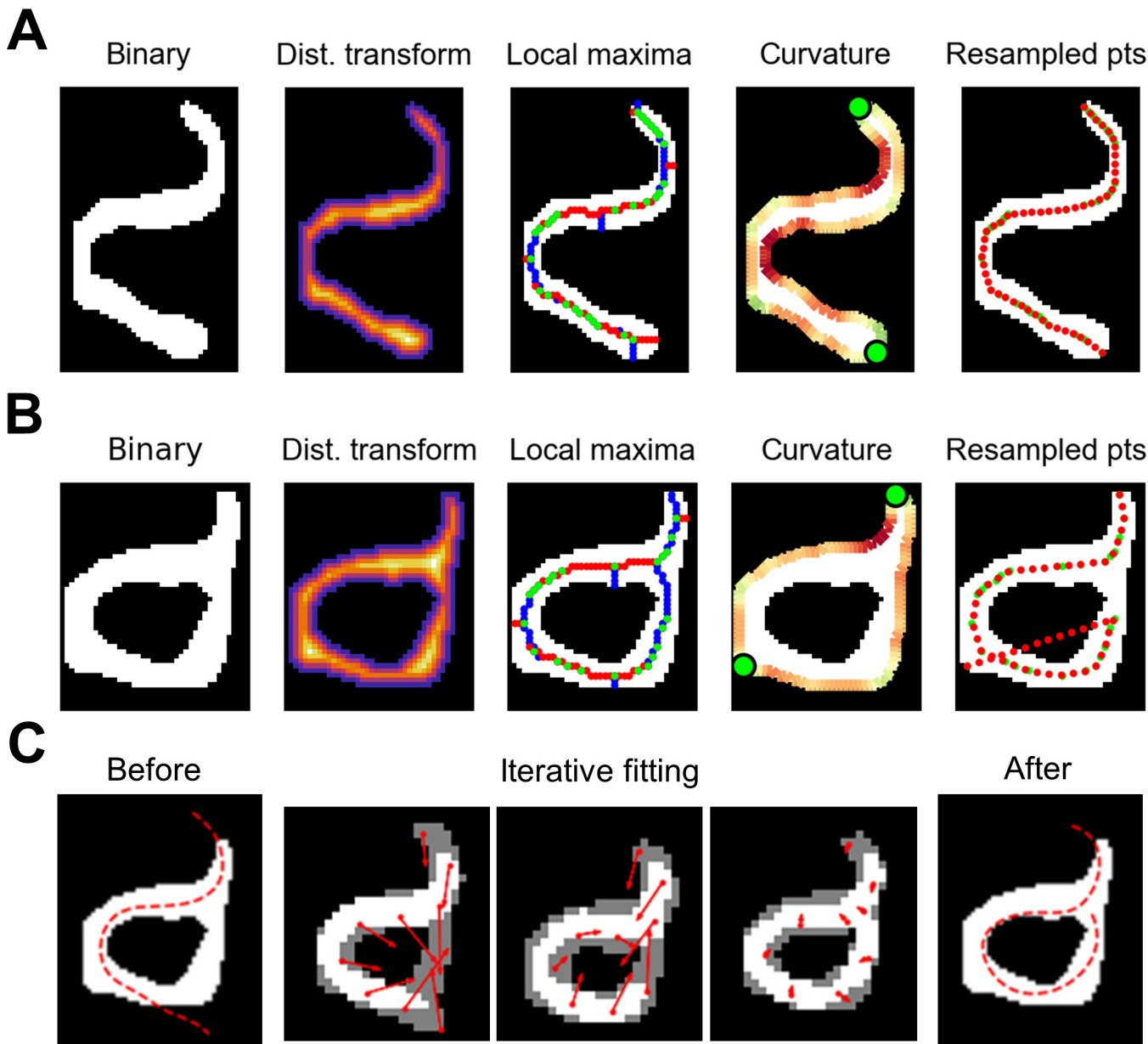

**Fig 2. Computing the centerline from a binary segmentation. (A)** Finding the centerline of a non-self-overlapping worm. (*left*) The binary segmentation is converted into a distance transform (*center left*). Warmer colors indicate larger values. Local maxima of the distance transform (*center*) with maxima in the vertical direction only in red, maxima in the horizontal direction only in blue, and maxima in both directions in green. Curvature of the perimeter (*center right*): red regions are concave, green regions are convex, and bright green circles indicate maximally convex points. (*right*) The resulting centerline points (red circles) based on co-localized local maxima and convex maxima (green). **(B)** The same procedure applied to a self-overlapping worm resulting in an erroneous centerline. **(C)** Correction of the erroneous centerline using a deformable model initialized based on a correct centerline from an adjacent frame (*left*). Points on the deformable model are pulled toward uncovered areas of the segmentation that caused the erroneous centerline (*center three panels*). Red arrows represent the direction and magnitude of pull on each model point. (*right*) The corrected centerline.

and nictation duration, the average duration of a nictation bout [5]. Because longer nictation bouts often begin before or end after a worm track, we are likely undersampling long bouts. Therefore, we calculated stopping rate (SR), the rate at which nictating worms stop nictating, instead of nictation duration, to better describe the behavior based on worm tracks.

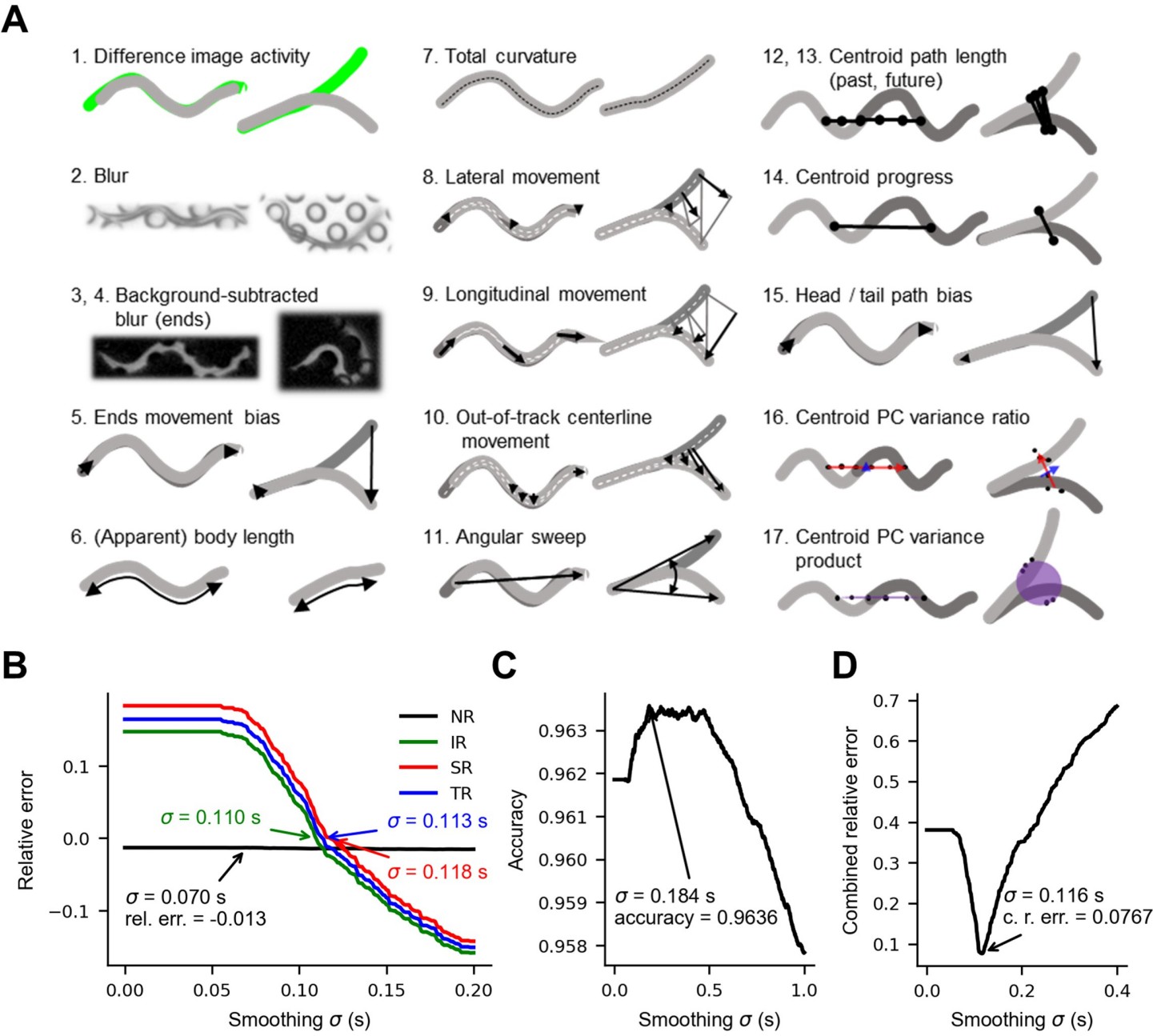

**Fig 3. Feature-based detection of nictation. (A)** Cartoon schematics of the 17 features used to score nictation, each showing a recumbent worm (left) and nictating worm (right). **(B)** The effect of smoothing model output probabilities on the accuracy of common nictation metrics and overall transition rate. *Relative error* is the value calculated from the smoothed model probabilities divided by the value calculated from manual scores, minus one (equal to zero when there is no error). **(C)** The effect of smoothing model output probabilities on computer score accuracy on the validation data. **(D)** The effect of smoothing model output probabilities on the sum of the relative error of NR, IR, and SR as well as 1—accuracy. In panels B, C, and D, the accuracies shown are averages computed from the validation set accuracies during five-fold cross validation. PC = principle component, σ = standard deviation used in smoothing model probabilities, rel. err. = relative error, NR = nictation ratio, IR = initiation rate, SR = stopping rate, TR = (overall) transition rate, c. r. err. = combined relative error.

Examination of the scored behavior shows that the neural network behavioral scores contain far more behavioral transitions (nictation initiations and terminations) than the manual scores (1868 versus 1114 transitions in the test set). As a result, IR and SR calculated from the raw computer scores were substantially higher than IR and SR calculated from human scores.

We tried several methods to correct these errors, including applying Gaussian smoothing to the probabilities output by the model, eliminating short behavioral bouts, using the neural network scores as observations in a hidden Markov model, and using the neural network probabilities as emission probabilities in a hidden Markov model (**S5 Fig**). Of these, Gaussian smoothing is the simplest and resulted in the best performance overall (**Fig 3B–3D**). Therefore, we used Gaussian smoothing for the remainder of this study to eliminate short, often erroneous bouts. Smoothing reduced both IR and SR while having a minimal effect on nictation ratio, which was slightly below the ground truth value (**Fig 3B**). Overall accuracy also increased from 96.00% to a maximum of 96.36% when smoothing with a σ = 0.184 s Gaussian kernel (**Fig 3C**). By scoring the test dataset with different subsets of features scrambled by randomly permuting them, we found that accuracy decreased from 94.1% to 47.8% if all features were scrambled, to 86.4% if only the 16 primary features were scrambled, and to 53.2% if all features derived from multiple worm-frames were scrambled, suggesting that information from multiple worm-frames is important for scoring accuracy.

We noticed that slightly different smoothing amounts were optimal for IR and SR. This may arise when there is an error in nictation ratio. For example, if the correct number of nictation bouts, and therefore transitions, are detected, but they are too short (NR too low), then IR will be lower than the true value, and SR higher than the true value because the denominators of those two metrics would err in opposite directions. This is indeed what we see in our nictation dataset, for which the model NR is slightly too low: when enough smoothing is applied to correct the overall transition rate, SR is too high and IR is too low (**Fig 3B**).

In subsequent *C. elegans* analyses, we applied smoothing with a Gaussian kernel of σ = 0.116 s when comparing the model to human scorers, and a Gaussian kernel of σ = 0.193 s when scoring the full *C. elegans* dataset. These σ values were chosen to optimize for a combination of per frame accuracy and the nictation metrics NR, IR, and SR when five-fold cross-validation was performed on just the training set or the combined training and test sets, respectively (**Figs 3D and S6**).

## Machine learning model validation and comparison with human judges

While human scoring is often considered the "gold standard" or "ground truth", the scores of trained experts using the same rubric to score the same videos of animal behavior can differ substantially [41]. To determine whether a neural network model is more consistent with the trainer than with different humans, we trained a model on the entirety of the training video and compared its performance on the testing video to that of two human scorers, humans 2 and 3, who also scored the testing video using the same definition of nictation as the trainer.

Side by side examination of the scores from the three humans and the model show broad agreement in terms of the temporal location of nictation bouts (**Fig 4A**). In terms of accurate nictation metric values, the smoothed model output, with smoothing optimized based on the training dataset only, outperformed the raw model output, and we refer to this output in the following comparison. In terms of accuracy, the model agreed more closely with its trainer than with the other humans, agreeing with the trainer 94.7% of the time, compared to 93.5% and 89.4% agreement for humans 2 and 3, respectively (**Fig 4B**). For nictation ratio, the computer's value of 0.420 also came closer to the trainer's value of 0.422, while humans 2 and 3 both scored more worm frames as nictation, with NRs of 0.447 and 0.491, respectively. For the transition rates, the errors were again in opposite directions, with the computer scoring both IR and SR too high (0.101 Hz and 0.138 Hz, respectively, versus 0.073 Hz and 0.094 Hz trainer values. Meanwhile, humans 2 and 3 both scored IR at 0.050 Hz and SR at 0.058 Hz and 0.049 Hz, respectively, lower than the trainer values (**Fig 4B**). These results demonstrate that our

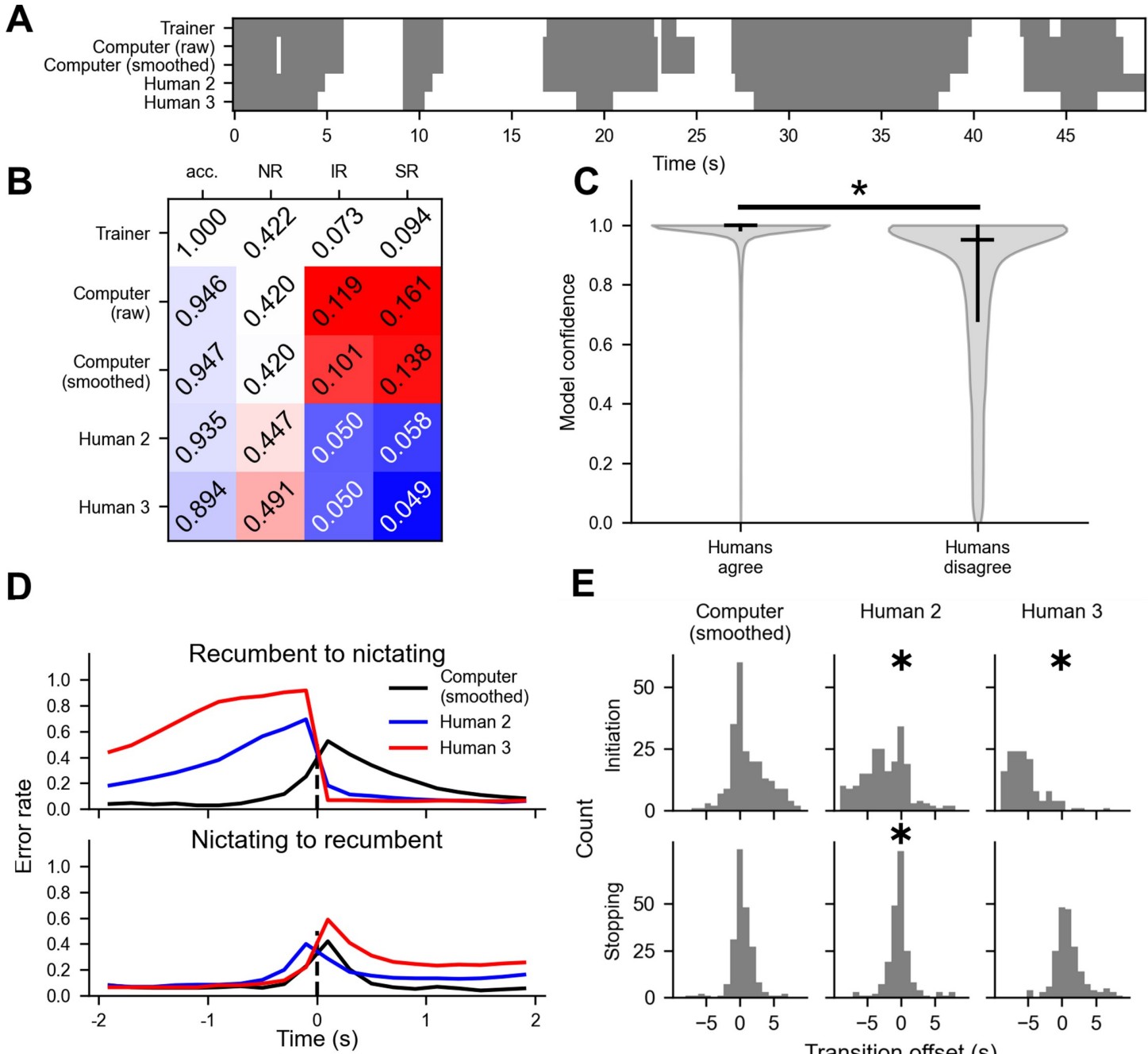

**Fig 4. Comparing computer model and human nictation scores. (A)** Ethogram of a worm track showing behavior according the trainer (top row), raw model scores (second row), smoothed model scores (third row), human 2 (fourth row), and human 3 (bottom row). Gray is recumbence, white is nictation. **(B)** Scoring accuracy and common nictation metrics computed from the scores of the human trainer, the raw model scores, the smoothed model scores, and the two other human scorers. Red and blue indicate that the scored value is higher or lower than the ground truth value, respectively, and lighter shades indicate closeness to the human trainer values. **(C)** Violin plot of the confidence of the computer model–calculated as the difference between the probability of the behavior scored and the second highest probability–for worm frames in which all three humans agreed (n = 59743), or in which at least one human disagreed (n = 8965). The black horizontal line indicates the median and the vertical black line indicates the first and third quartiles. The groups are significantly different (p = 0.000, Wilcoxon rank-sum test). **(D)** Accuracy of the smoothed computer model scores and the scores of humans 2 and 3 near behavioral transitions scored by the trainer. Scores were compared to those of the trainer whenever the trainer scored a transition preceded and followed by at least two seconds of uninterrupted behavior. **(E)** The time offset of nictation initiation and termination of the computer model and human scorers 2 and 3 relative to the trainer. Only transitions occurring within two seconds of the trainer-scored transition are shown, and only if there was only one of the relevant type of transition scored in the that time window. ∗ denotes a median significantly different from zero (binomial test, p < 0.05 after Bonferroni correction).

model can score nictation with accuracy similar to or better than a human scorer (when compared to the trainer's scores), but that behavioral transition rates may vary between scorers and scorer types.

Previous work comparing multiple human scorers with an algorithm suggests that certain behavioral bouts are inherently more difficult to score for both a computer model and humans [41]. We reasoned that difficult-to-score bouts might be indicated by disagreement among humans and low confidence in model predictions. To avoid complicated issues in defining individual bouts of nictation across scorers, we tested this idea on a per-worm-frame basis, using the margin of model probability for the scored behavior (probability of the scored behavior minus the probability of the second most likely behavior) as a measure of model confidence. Indeed, the model showed significantly higher confidence (p = 0.000) on frames where the three humans agreed (median probability margin = 0.999, IQR = 0.986–0.999) than when at least one did not agree with the other two (median probability margin = 0.951, IQR = 0.682–0.995) (**Fig 4C**). Inspection of the behavioral scores suggests that these low-confidence scores are concentrated near the initiation and termination of bouts. Indeed, humans 2 and 3 show a greater propensity for error just prior to nictation initiation or after nictation termination (**Fig 4D**), consistent with the tendency of humans 2 and 3 to score nictation bouts as beginning significantly earlier than the trainer, and with human 2 also scoring nictation bouts as ending significantly earlier than the trainer (p < 0.05, two-sided binomial test with Bonferroni correction) (**Fig 4E**). Taken together, our results demonstrate that both human scorers and our model make more errors near behavioral transitions, but that human scorers may be more prone to making systematic errors in the timing of behavioral bouts.

## Nictation behavior arises with SDS resistance in liquid-cultured *C. elegans*

While the formation of *C. elegans* dauers used in studies of nictation has usually been induced by starvation or the addition of exogenous dauer pheromone [14, 16, 17], high density, low food liquid culture conditions can be used to grow *C. elegans* dauers without added pheromone [42]. To determine how nictation behavior arises in *C. elegans* reared in such conditions, we recorded videos of animals on a microdirt arena at 2, 2.5, 3, 4, 6, 10, 14, 21, and 28 days after feeding arrested L1s and used our automated behavior scoring pipeline to track them and score their nictation (**Fig 5A**). We observed an overall trend of increasing nictation, with significant differences in NR between day 2 (the earliest timepoint assayed) and days 6 and 21 (Dunn's test p < 0.05 after Bonferroni correction for 36 comparisons) (**Fig 5B**). We also observed a dip in NR at day 10 confirmed by manual scoring of a subset of the data (**S7 Fig**).

Previous studies of dauer development in liquid media have found that dauer formation occurs over the course of five or six days [42, 43]. A characteristic of dauers that distinguishes them from other life stages is increased resistance to SDS [2]. Our liquid culture animals increase in SDS resistance over the course of the six days after feeding (**Fig 5C**), while length changes little (**S8 Fig**), staying within the reported range for *C. elegans* dauers [44], suggesting that the increase in nictation coincides with the transition of these animals from L2d pre-dauers to dauer larvae. These findings demonstrate our ability to automate the scoring of nictation in *C. elegans* dauers.

## Four minutes of video suffice to quantify population nictation behavior

Experimental throughput can be limited by long recording times and large datasets, so we asked how much data collection is necessary to describe how NR changes with development. NR computed from as little as the first five seconds of each video shows the same general trends over age as NR computed from the full video (**Fig 6A**), with slightly higher NRs due to

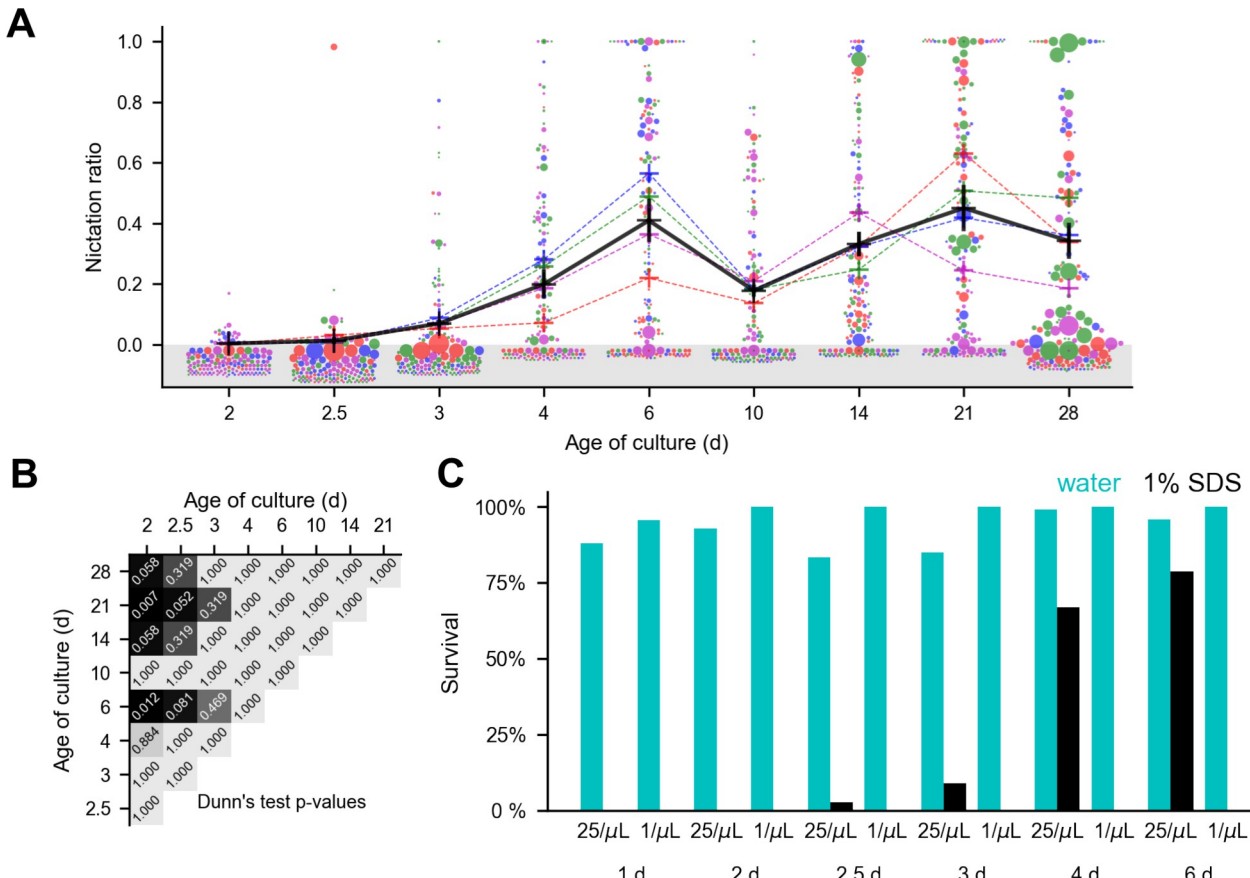

**Fig 5. *C. elegans* nictation ratio depends on time in culture. (A)** Nictation ratio of liquid culture dauers assayed at different times after refeeding arrested L1s. Colored crosses represent video averages and black crosses represent timepoint averages. Circles represent values calculated from individual worm tracks, with the area proportional to the duration of the worm track and separate videos indicated by color. Only 150 worm track values are shown per timepoint due to space limitations, and all circles positioned below zero (gray shaded area) represent zero. **(B)** Dunn's test p-values of video average NRs from panel A after Bonferroni correction. **(C)** Survival after a 1 h exposure to water (cyan) or 1% SDS (black) of liquid culture *C. elegans* grown at 25 or 1 worm per μL and tested at various times after refeeding.

decreasing NR over the course of the 30 min videos (**S9 Fig**). Because undersampling can result in noisier observations, we asked when continued recording no longer decreases sampling noise. We computed the relative variance of video-wise NRs at each culture timepoint using different video time cutoffs. We saw no further decrease in relative variance after 3 min 57 s of recording (**Fig 6B**). This indicates that while we recorded for 30 min, for measuring NR no further benefit was achieved by recording longer than about 4 min.

### Measuring *Steinernema carpocapsae* nictation in the absence and presence of potential hosts

To test the effectiveness of our nictation scoring pipeline on EPNs, we recorded videos of *S. carpocapsae* IJs on microdirt arenas. We used manually-scored ground truth data to train a second neural network model as well as to optimize Gaussian smoothing amount for this species (**S10 Fig**). In initial testing, nictation was rarely observed. Because cues such as $CO_2$, air movement, and the odor of potential hosts have been observed to induce nictation and other host-finding behavior in some EPN species [10, 45], we recorded 15 min of baseline behavior and then added three *G. mellonella* larvae to a small cage inside the Petri dish containing the

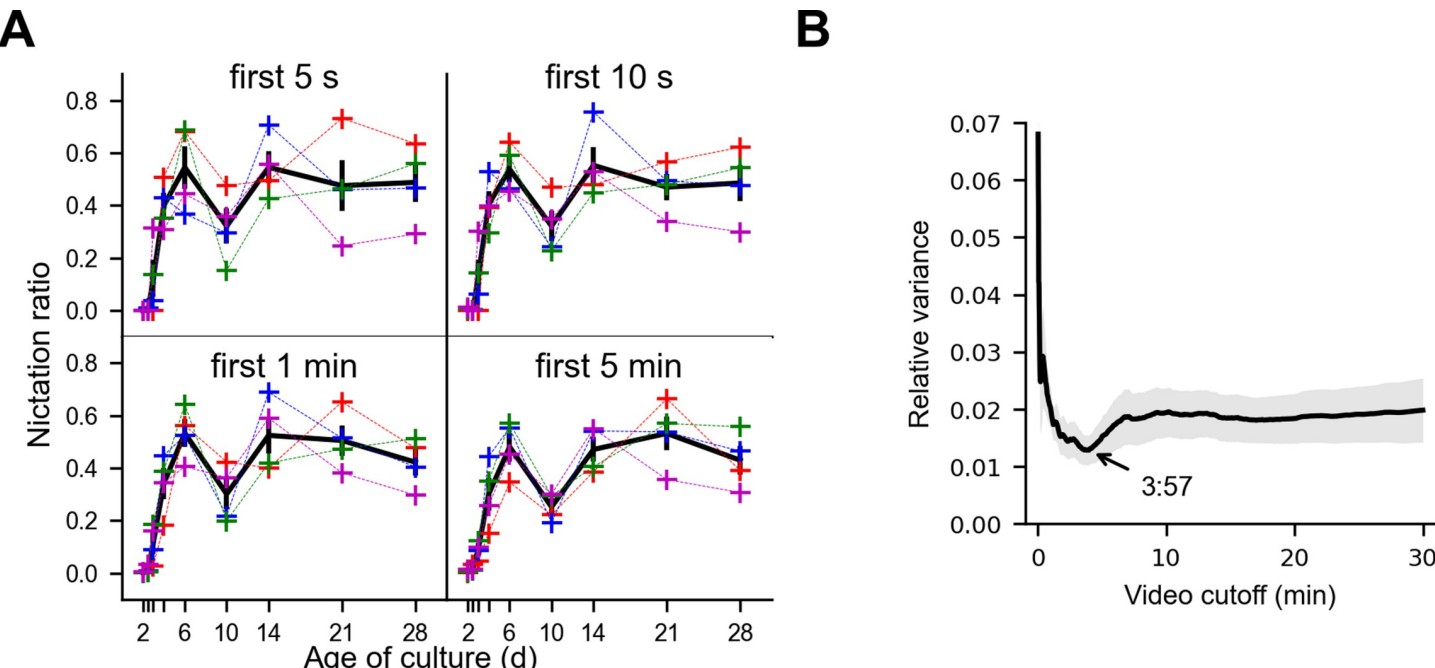

**Fig 6. Four minutes of video suffice to describe nictation ratio.** (A) Nictation ratio calculated using the first 5 s, 10 s, 1 min, and 5 min of data from each video. Colored crosses indicate video NRs, black indicates the average of the video values, and black vertical bars show ±SEM. (B) Relative variance (variance/mean of NR from all four videos at each timepoint) of the nictation ratios calculated using the first portion of the video. The gray area represents ±SEM (of the relative variances). The cutoff resulting in the lowest relative variance, 3:57, is indicated.

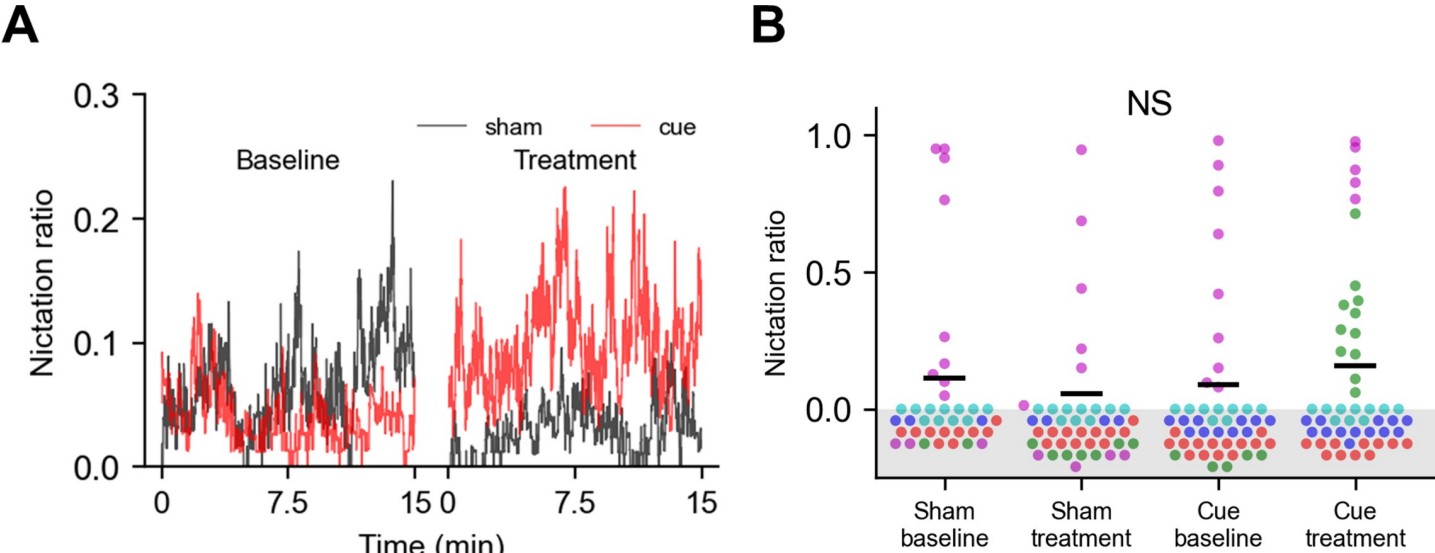

**Fig 7. _Steinernema carpocapsae_ nictation in the presence of hosts.** (A) Average nictation ratio of baseline and treatment videos over the course of the 15 min videos. (B) Nictation ratio calculated from individual worm tracks (dots), each of 60 s duration and representing a unique worm. Colors correspond to each of the five replicates. The groups were not significantly different (Kruskal-Wallis test, p = 0.15). All worm tracks include the 3893[rd] frame of their respective videos, and were truncated to 60 s prior to calculating NR.

microdirt arena. These videos, together with controls in which we opened and resealed the Petri dish but did not add *G. mellonella* (sham), allowed us to use our pipeline, retrained for *S. carpocapsae* (see Methods), to evaluate the effect of host cues on EPN nictation behavior.

Observing NR over time in animals following exposure to *G. mellonella* larvae suggests a small, but variable increase in nictation (**Fig 7A**). To test this statistically, we wanted to compare the behavior of individual worms from these videos. To do this, we limited our analysis to worm tracks that met two criteria: (1) they were at least 1 min in duration, truncated to 1 min, to provide a consistent snapshot of behavior, and (2) they were tracked at frame 3893, so we could be sure we were looking at a set of unique animals and controlling for time spent on the arena. Frame 3893 was chosen because all videos were represented by at least two tracks of sufficient duration at that frame, and because among such frames, this was the one where the greatest total number of qualifying worm tracks coincided (**S11 Fig**). The increase in worm-wise NR did not reach statistical significance (Kruskal-Wallis test, p = 0.15). Furthermore, observation of the worm-wise data suggests trial-to-trial variation, with one trial containing tracks of elevated nictation in all four videos, and another showing nictation only after addition of the *G. mellonella* (**Fig 7B**). These results demonstrate the ability of our pipeline to score nictation in an EPN species, and efficiently score large amounts of data, exposing considerable trial-to-trial variation in behavior.

## Discussion

In this study we used a Mask R-CNN to detect and segment free-living and entomopathogenic nematodes on a textured background designed to promote nictation behavior. We converted these segmentations to centerline coordinates using a novel, ridgeline-based approach, or, if that failed, a deformable model. Using features mostly calculated from these coordinates and ground truth manual behavior labels, we trained a computer model to classify nematode behavior, focusing on scoring the host-finding / phoretic nictation behavior of the dispersal stage of these animals.

Mask R-CNN was less sensitive to the choice of thresholding parameters than classical intensity-based segmentation, likely reducing the risk associated with reusing the same parameters across multiple videos that may differ slightly in contrast. The reason for this advantage is the relatively high contrast grayscale images output by the Mask R-CNN compared to those resulting from background subtraction in intensity segmentation (**S2 Fig**). While we achieved similar segmentation quality using classical intensity-based segmentation, the size of the gaps in the background-subtracted worm images was constrained by the width of the post edges, so our success may not be repeatable on other textured backgrounds with larger background features. Mask R-CNN has been used to segment objects, including partially-obscured objects, on a variety of complex backgrounds [27]. Therefore, we believe the advantage of Mask R-CNN may be greater in situations where the background is not as regular or contains larger features creating gaps that cannot be easily smoothed-over.

Converting two-dimensional worm segmentations into centerline coordinates has been a longstanding problem with many imperfect solutions published. Most methods use simple morphological thinning or curvature of the perimeter (*e.g.* Leifer *et al*. 2011; Yemini *et al*. 2013), but problematic postures involving self-overlap and collisions have been the focus of entire research papers [34, 38], including one leading to a commercial software package (WormLab®), and another utilizing crowd sourcing [46]. Our ridgeline method has the advantage of not requiring the pruning of "split ends", but does fail in some cases. While we were able to fix many of these using a deformable model, resulting in a substantial improvement in scoring accuracy, the deformable model is relatively complex and requires a

temporally adjacent accurate centerline for initialization. An algorithm is under development that can take an image of a population of *C. elegans* and output centerline splines directly, potentially bypassing this problem in the near future [47]. While a few of our features, like blur and difference image activity, are calculated based on the segmentation and raw image rather than the centerline, the bilateral symmetry of the worm makes going from a centerline spline to a 2D mask relatively straightforward.

To convert feature values into behavior labels, we tested combinations of commonly-used machine learning algorithms and scaling methods. Several of these combinations performed similarly, with accuracy around 90–95%, and, with most model types, the effect of scaling was minimal. This suggests that another factor is limiting performance. For example, scoring or tracking errors in the training data could interfere with training by providing "confusing" examples to learn on, and disrupt inferencing by providing features that do not accurately reflect the true position and movement of the worm. A human, on the other hand, views the video directly and is not dependent on accurate feature calculation. The lack of a performance boost from feature scaling, which is standard practice in the field, was not entirely unexpected for some model types. Algorithms based on distance (in feature space), such as k nearest neighbors, benefit from feature scaling because they prevent features with larger values from dominating, whereas tree-based algorithms like random forest should not benefit and indeed did not in our hands. Another reason that we saw limited benefit from scaling may be the fact that, when possible, our features were calculated in real units, and our subjects were relatively uniform in size.

Two things that improved accuracy were the inclusion of features calculated from multiple frames and smoothing, which blended together the behavior probabilities from nearby frames. This suggests that information from more than one frame may help to score behavior accurately in some cases. This is consistent with literature showing that even human scorers benefit from 5–7 frames worth of video in order to accurately classify human behaviors [48].

For the most part, the model in this study performed more like its trainer than two other human scorers. This is consistent with earlier findings that expert human scorers sometimes disagree when scoring behavior [41]. Mistakes tended to occur more often during behavioral transitions, which may not be as well-separated in feature space. Furthermore, worm-frames for which the three humans did not agree on a behavior are also scored less confidently by the model, as indicated by the margin of probability of the scored state. Unlike the model, however, the humans also made systematic timing errors. For example, both tended to score initiations too early. Such errors could be caused by differing interpretations of the scoring instructions, or even inattentiveness and fatigue.

An area where neither the model nor the human scorers were very consistent with the trainer was in scoring behavioral transition rates based on frame-by-frame scores. We tried several methods to tackle this problem of scoring "noise", but none outperformed simple Gaussian smoothing of the model output probabilities. Furthermore, error in the overall NR could cause the amount of smoothing required to optimize initiation rate and stopping rate to be different. Unfortunately, there seems to be a difference in the amount of smoothing needed to correct the transition rate in different videos, evidenced by the difference in optimal smoothing sigma when optimization was performed on only the training set or on the combined training and test sets of *C. elegans*, suggesting that these metrics may be highly sensitive to recording conditions.

Manual scoring of animal behavior continues to be quite common despite the rise of capable machine learning algorithms [36]. Therefore, while we do not anticipate that automated scoring will fully replace human inspection, we believe it will become increasingly important in large studies where consistency is paramount, such as genetic and pharmacological screens.

Other possible applications include quantifying the effects on nictation of the application of a variety of *in vivo* manipulations such as targeted mutations and gene overexpression and knockdown, as well as systematic studies of the effect of environmental conditions, such as temperature, humidity, and illumination, on nictation. Our data also complement previous work showing variability between different human scorers using the exact same instructions [41], whereas a computer model should perform more consistently provided the data are also consistent. Furthermore, certain rare, unusual behaviors like, in this study, "tail wagging", may be difficult to discover except by a human observing a large amount of video.

In conclusion, we have created a pipeline for deep learning image segmentation and feature-based scoring of animal behavior. While we have demonstrated its utility for studying a specific behavior–nematode nictation–the same approach could, in principle, be used to score other nematode behaviors, or even the behavior of other rod-shaped animals like zebrafish and *Drosophila* larvae (*i.e.* animals whose posture can be reasonably represented as a centerline spline). To do this would require only retraining of the Mask R-CNN and behavior classifier using manually-annotated video frames and behavior, respectively.

## Materials and methods

### Buffer recipes

**Phosphate-buffered saline (PBS).**   154 mM NaCl, 7.8 mM $Na_2HPO_4$, 1.6 mM $NaH_2PO_4$ (adjusted to pH 7.4)

**S-basal.**   5.7 mM $KH_2PO_4$, 44.1 mM $K_2HPO_4$, 100 mM NaCl (autoclaved)

**S-complete.**   5.2 mM $KH_2PO_4$, 40.1 mM $K_2HPO_4$, 90.9 mM NaCl, 9.1 mM K citrate, 27.3 mM $CaCl_2$, 27.3 mM $MgSO_4$, 4.5 mg/L cholesterol, 0.09% ethanol (cholesterol vehicle), 0.48X penicillin-streptomycin-neomycin (Gibco 15640055), 48 units / L nystatin (Sigma N1638), 20.1 μM $Na_2EDTA$, 9.0 μM $FeSO_4$, 3.7 μM $MnCl_2$, 3.7 μM $ZnSO_4$, 0.6 μM $CuSO_4$ S-basal: 5.7 mM $KH_2PO_4$, 44.1 mM $K_2HPO_4$, 100 mM NaCl (autoclaved)

### *C. elegans* strains and maintenance

We used Bristol N2 maintained at 20˚C according to established protocols [12]. For general maintenance and experiments, we reared *C. elegans* on 90 mm Petri dishes containing Nematode Growth Medium (NGM) agar seeded with *E. coli* OP50 bacteria for food. Our formulation of NGM is the same as described [12], except we use 5.0 g/L bacto-peptone.

### Synchronized *C. elegans* liquid culture

To generate synchronized dauer cultures, we picked five gravid adults onto each of twelve 90 mm OP50-seeded NGM dishes and incubated them at 20˚C for four days. Then we used S-basal to wash the worms into a 15 mL conical tube and isolated their eggs using hypochlorite treatment [12] and incubated them overnight at 20˚C on a rotating stand to prevent settling. After 19 h, we measured the density of L1s by counting the number of worms in three 20˚L drops and adjusted the volume so that about 38750 (25 worm/˚L final concentration) or 1550 (1 worm/˚L final concentration) worms remained in the tube. We pelleted these and reduced the supernatant to 50˚L. We next resuspended the L1s in 1.5 mL S-complete supplemented with *E. coli* HB101 at an OD600 of 1.95 and incubated them at 20˚C on a rotating stand until needed. For nictation experiments, dauers were drawn serially from the same cultures, and for SDS resistance assays, one culture was used per assay.

## *S. carpocapsae* strains and maintenance

We used *S. carpocapsae* strain All, obtained from Johnathan Dalzell (Queen's University, Belfast). To generate IJs for experiments, we pipetted IJs from a previous White trap [49] onto PBS-moistened filter paper in a 60 mm Petri dish. We also transferred five to ten live *Galleria mellonella* larvae onto the filter paper, and parafilmed and incubated the Petri dish at 23˚C for two to four days. After incubation, we transferred infected cadavers to a White trap. This consisted of PBS-moistened filter paper covering the upper surface of the lid of a 90 mm Petri dish, which was itself situated inside another 150 mm Petri dish partially filled with PBS such that the lid formed an island. After 14 to 21 days, IJs emerged and entered the PBS.

## Imaging setup

The imaging system comprised a grayscale machine vision camera (Imaging Source DMK27AUP031), 50 mm lens (Kowa LM50JC10M), and clear acrylic stage with brightfield illumination provided by a square array of white LEDs (Rosco LitePad CCT) covered with a layer of diffusing paper. The incidence angle of illuminating light was constrained with a 22 mm inner diameter lens tube and two perpendicular layers of light film (Edmund Optics 52–385) taped to the underside of the stage. The field of view was approximately 5.0 x 3.7 mm with a resolution of 2560 x 1920 pixels. We rescaled the videos to 1280 x 690 pixels to avoid memory limitations during processing.

## Design and fabrication of microdirt arenas

The microdirt arenas have been described previously [5, 14]. Each arena consists of an approximately 3 mm thick slab of molded agar with a rectilinear grid of cylindrical posts of 50˚m diameter and 25˚m height spaced 75˚m apart, center to center, covering the top surface. We cut the arenas into 15 mm squares for use in experiments.

To make the arena mold, first a positive mold was created by photolithography of a layer of photoresist (SU-8 2008, Microchem) on a 4 inch silicon wafer [50]. We then poured polydimethylsiloxane (PDMS) (Silgard 184, Dow-Corning) onto this positive mold, degassed the PDMS under vacuum, and baked the PDMS at 65˚C for 1 h to create a negative mold. We used the negative mold to create the microdirt arenas. Several hours prior to an experiment, we autoclaved a 4% solution of agar in RO water and poured it onto the negative mold to the desired thickness (3 mm). Immediately after pouring, we used the edge of a glass slide to remove air bubbles from the post holes on the negative mold.

## Nictation assays

We performed all nictation assays in a climate-controlled room (set at 20–21˚C for *C. elegans* and 23–24˚C for *S. carpocapsae* and 30–45% relative humidity for both). For *C. elegans*, we pipetted 75˚L of liquid culture suspension into a 1.5 mL microcentrifuge tube and allowed the worms to settle for 5 min. Meanwhile, we applied a thin layer of Tween 20 to the inside of the lid of a 90 mm Petri dish to prevent fogging during imaging, and placed a 15 x 15 mm arena inside. After 5 min, 10˚L of the settled worm suspension was pipetted onto the four corners of the arena. The pipette tip used had about 3 mm snipped off to widen the orifice and allow the worms to flow freely during pipetting. We allowed the arena with worms to dry for 10 min with the lid off. After drying, we added two pieces of laboratory wipes moistened with RO water alongside, but not touching, the arena, and parafilmed the Petri dish to minimize drying and shrinking of the arena during imaging. We transferred the Petri dish to the imaging rig. In keeping with previous, manual assays wherein only animals that had begun to move were

scored [5, 14, 51, 52], we positioned the arena so that the field of view was roughly centered and only animals that had moved inward from the corners could be recorded. We loaded *S. carpocapsae* IJs the same way, except that a variable volume of culture (100–170˚L), chosen to contain about 350 IJs, was used in the settling step. We recorded videos at 5 fps starting immediately after loading was complete unless otherwise specified.

## Evaluation of segmentation by intensity and Mask R-CNN

We wrote custom image processing scripts in Python. We initially segmented worms based on intensity or using a Mask R-CNN [27]. For intensity-based segmentation, we subtracted the max-merge background from each frame, smoothed it with a Gaussian filter of $0 \leq \sigma \leq 39$ µm (1 µm increments converted into pixel units), binarized it at a grayscale threshold of $5 \leq t \leq 124$ (increments of 1), and kept all above-threshold pixels. For Mask R-CNN-based segmentation, we used the PyTorch library [53] to fine tune a Mask R-CNN with a ResNet-50 backbone [54] on a training set consisting of 45 frames containing 1313 manually-segmented worms on a microdirt arena using a PC with a GPU (Nvidia GeForce RTX 2060). We applied the fine-tuned Mask R-CNNs to individual video frames and used max-merge to combine the masks of objects detected with a confidence of at least 0.7. We segmented the resulting grayscale image in a manner identical to intensity-based segmentation of a background-subtracted frame, except we tested Gaussian smoothing of $0 \leq \sigma \leq 69$ µm and grayscale thresholds of $25 \leq t \leq 249$. For evaluation, we compared these segmentations to manual annotations. We used detection (yes or no), IoU, number of gaps in the segmentation, and centerline RMSD as evaluation criteria. A worm was considered detected if any part of any ROI overlapped any manually-annotated pixel of that worm. The IoU was calculated between manually-annotated worms and all overlapping ROIs. If it was detected, the number of gaps in a manually annotated worm was equal to one less than the number of overlapping ROIs. The centerline RMSD was calculated by calculating the RMSD of the manually-drawn centerline and centerlines calculated from all overlapping ROIs (see centerline finding) in both possible orientations and taking the minimum value.

## Tracking

Tracking consisted of three steps: segmentation, centerline determination, and frame stitching, described below.

**Segmentation.** We used a Mask R-CNN, described above, to detect *C. elegans* dauers. For *S. carpocapsae* IJs, we used a similar Mask R-CNN fine-tuned on a training set consisting of 35 frames containing 653 manually-segmented IJs. For *C. elegans*, we used a smoothing $\sigma = 6.4$ µm and a grayscale threshold of 100. We excluded ROIs whose area was not between 3674 and 16533 µm$^2$, the typical area range of ROIs representing individual dauers. For *S. carpocapsae*, we used a grayscale threshold of 100 without smoothing and kept ROIs between 4000 and 22000 µm$^2$. For both species, we excluded ROIs that touched the edge of the video frame.

**Centerline determination.** First, we found the nose and tail tip (endpoints) of each ROI by finding two minima of the interior curvature of the ROI outline. Next, we took the distance transform of the ROI and found points that were local maxima in both the horizontal and vertical directions (ridge points). We determined the centerline by fitting a spline to the endpoints and ridge points and dividing it into 50 equally-spaced points. This produced erroneous centerlines for worms in certain configurations such as omega turns. Centerlines of length $> 750$ µm, or containing angles sharper than 45 degrees, or that intersected themselves were flagged for fixing using a deformable model.

The deformable model consisted of nine centerline points initially positioned along the nearest (in time) non-flagged centerline from the same worm track (see frame stitching). These points are used to draw a binary moving ROI of area equal to the ROI whose centerline was flagged (target ROI). The moving ROI was adjusted to better overlap the target ROI in an iterative fitting process. In each iteration, an overlap of the moving and target ROIs is generated and an attractive "force" is calculated between each point and any non-overlapped pixels of the target ROI. These "forces" are applied separately to each model point to update its positions, and also combined to calculate a net force and "torque" to translate and rotate the entire model. Fitting was considered complete after 50 iterations if IoU > 0.6 and improvement had plateaued (no net improvement in fit over the previous 50 iterations), or if 300 iterations had elapsed. Worm frames in which the centerline still met flagging criteria after this process were excluded from analysis.

**Frame stitching.** We stitched worms detected in adjacent frames into worm tracks based on centroid proximity, matching the closest centroids first. We excluded any matches where the centroids in adjacent frames were more than 429 μm apart for *C. elegans* or more than 300 μm apart for *S. carpocapsae* to reduce identity switching within worm tracks. We used endpoint proximity to align centerlines from frame to frame. Worms tracks of duration less than an arbitrary threshold of 10 s were excluded from further analysis.

## Manual scoring of nictation

Human scorers used a GUI that allowed them to play, pause, reverse, or step frame by frame through videos of single tracked worms. Worms were scored as nictating if at least 1/5 of their total length was lifted off the substrate, otherwise they were considered to be recumbent. Worms were also scored as active or quiescent, but this distinction was not used in further analysis. Worm-frames in which behavior could not be scored for any reason, including tracking errors, were scored as censored.

## Automated scoring of nictation

For *C. elegans*, the training set consisted of 516 worm tracks from a single video from which 69312 manually-scored worm frames were used, of which 35198 were scored as recumbent, and 34114 were scored as nictating by the trainer. Frames scored as censored or containing flagged centerlines that could not be fixed using the deformable model were excluded, as were any other frames within two frames of these. The testing set consisted of 374 worm tracks from the same video with 68708 worm-frames, of which 39533 were scored as recumbent, 28831 were scored as nictating, and 344 were censored. Flagged centerlines that could not be fixed were excluded as in the training set, but censored frames were left in to better simulate performance on data that had not been manually-scored. We calculated the features shown in Fig 3A based on the tracking output. Min-max scaling of features, a neural network model, and Gaussian smoothing of the model output probabilities prior to extracting the final score for each worm frame were used based on performance in the model and scaling evaluation shown in S3 Fig. Prior to scoring the rest of the *C. elegans* dataset, we trained the model on both trainer-scored videos, excluding censored frames from the testing set.

We used a similar method for *S. carpocapsae*, except that the training set consisted of 175 worms tracks containing 32354 and 7898 worm frames scored as recumbent and nictating, respectively, and the test set consisted of 170 worm tracks containing 37676, 9662, and 69 scored as recumbent, nictating, and censored, respectively. Five-fold cross-validation was used to calculate the Gaussian smoothing sigma only, and a neural network model was trained on both videos before using it to score the remaining videos in the dataset.

## Smoothing model scores

We evaluated four methods to eliminate behavioral scoring noise in the raw model output. For all four methods, we tried a series of parameter values that changed the degree by which noise was reduced. We compared these methods by applying them to the models and datasets used in five-fold cross-validation of the neural network model.

**Gaussian smoothing.** We applied Gaussian smoothing in time, varying σ from 0 to 1 s, to the behavior probabilities output by the model. Frames were given the score of the most probable behavior after smoothing.

**Removing short bouts.** Proceeding through each worm track frame by frame, any bout of behavior whose duration was less than the minimum bout length was merged into the surrounding bout, regardless of behavior type. While we always proceeded from beginning to end, this procedure can yield slightly different results if implemented in reverse, for example, when there is rapid switching between behavioral states at the end of a longer bout of behavior.

**Viterbi state-based.** We treated the raw model scores as observations in a hidden Markov model with emission probabilities based on typical model accuracy. This performed poorly when using state transition probabilities calculated from manual scores, so we instead varied the transition probabilities, keeping the transition matrix bisymmetric. We varied the probability of remaining in the same state (what we call the *persistence probability*, $P_P$) between 0.45 and 1.00, with a complimentary probability of changing states, $P_C$. Initial behavior probabilities were based on the ratio of behavior observed in the manual scores of the training data, and reflected the NR. We then used the Viterbi algorithm to find the most likely sequence of behavioral scores based on the raw model scores and the parameters described above.

**Viterbi probability-based.** Instead of using the raw model observations, we used the model probabilities directly as the emission probabilities of observations in a hidden Markov model and ran a modified version of the Viterbi algorithm to find the most likely sequence of behavior. As before, we varied the persistence probability to modulate the denoising effect.

## Statistics and plotting

Statistical tests and significance cutoffs are described in the figure captions and Results. Bonferroni corrections were used for multiple comparisons and non-parametric tests were used when the data did not follow a normal distribution according to the Anderson-Darling test. All statistical tests were implemented using the SciPy Python library [55]. A cutoff of $\alpha = 0.05$ was used for statistical significance throughout the manuscript.

## Supporting information

**S1 Fig. Segmentation parameter optimization. (A)** Mean IoU of manual and machine vision (computer) ROIs for a range of grayscale thresholds $t$ and smoothing sigmas $\sigma$. All computer ROIs that overlapped the manual ROI were combined for this calculation. If no computer ROI overlapped a manual ROI, the IoU was zero. The highest values for intensity- and Mask R-CNN-based segmentations are labeled. **(B)** Mean number of gaps per segmentation. The number of gaps for each manual ROI was one minus the number of computer ROIs overlapping it. Only manual ROIs that overlapped with computer ROIs were included. The highest mean IoU for $t$ and $\sigma$ combinations resulting in zero gaps is shown for each segmentation method. **(C)** Mean centerline RMSD for centerlines calculated from computer ROIs versus manually-drawn centerlines. The best (lowest) RMSD was used in cases where multiple computer ROIs overlapped a manual ROI, and a manual ROI was excluded if no computer ROI overlapped with it. The $t$ and $\sigma$ combination resulting in the lowest centerline RMSD for each

segmentation method is labeled with an "x". **(D)** Detection rate of manually drawn ROIs using the two machine vision methods and different combinations of $t$ and $\sigma$. A manually-drawn ROI was considered detected if it overlapped with at least one computer-drawn ROI.
(TIF)

**S2 Fig. False color images showing contrast of a background-subtracted difference image and the Mask R-CNN output.** (top) Cropped image of a *C. elegans* dauer after contrast enhancement by image stabilization followed by background subtraction. (bottom) Cropped image of the Mask R-CNN output of the same worm from the same frame of the unstabilized version of the video.
(TIF)

**S3 Fig. Model and feature scaling performance in five-fold cross validation. (A)** Accuracy on the training, validation, and test data of different algorithm–scaling combinations. All values are means from five-fold cross validation. The training and validation sets consisted of 4/5 and 1/5 of the worm tracks from the training video (74777 worm-frames or 4.15 h of manually-scored behavior—48.4% nictating, 51.2% recumbent, 0.4% censored—divided into 516 worm tracks), and the test set consisted of all the worm tracks from a separate testing video (72702 worm-frames or 4.04 h of manually-scored behavior—41.8% nictating, 57.5% recumbent, 0.7% censored–divided into 374 worm-tracks). **(B)** Training and inference times using different algorithm and scaling combinations (rows are different algorithms, columns are different feature scaling methods with training and inference times in adjacent columns). Training was done on a PC with an Intel Core i5-8265U (1.6 GHz) and 8 Gb RAM.
(TIF)

**S4 Fig. PCA plot of recumbence and nictation.** Red and blue dots represent worm-frames scored as nictation and recumbence, respectively, by the model and trainer, and bright green dots represent worm frames scored differently by the model and trainer. A subset consisting of 1% of worm-frames, sampled uniformly in time, is shown for clarity. The small cluster (arrow), is the result of a single worm, not a cluster of similar behavior from different worms.
(TIF)

**S5 Fig. Alternatives to Gaussian smoothing. (A-C)** Accuracy of common nictation metrics, scoring accuracy, and combined metric and scoring accuracy of model scores after removing short behavioral bouts. **(D-F)** Accuracy of common nictation metrics, scoring accuracy, and combined metric and scoring accuracy of model scores corrected using the Viterbi algorithm with raw model scores as observable states. **(G-I)** Accuracy of common nictation metrics, scoring accuracy, and combined metric and scoring accuracy of model scores corrected using the Viterbi algorithm with model probabilities as emission probabilities of the hidden Markov model. *Persistence probability* refers to the probabilities along the diagonal of the transition matrix in the hidden Markov model. All values are means from five-fold cross validation. The training and validation sets consisted of 4/5 and 1/5 of the worm tracks from the training video. NR = nictation ratio, IR = initiation rate, SR = stopping rate, TR = transition rate, p.p. = persistence probability, c. r. error = combined relative error.
(TIF)

**S6 Fig. Effect of Gaussian smoothing on nictation metric accuracy–combined *C. elegans* training and test sets. (A)** The effect of smoothing model output probabilities on the accuracy of common nictation metrics and overall transition rate. *Relative error* is the value calculated from the smoothed model probabilities divided by the value calculated from manual scores minus one. **(B)** The effect of smoothing model output probabilities on computer score accuracy

on the validation data. **(C)** The effect of smoothing model output probabilities on the sum of the relative error of NR, IR, SR and 1—accuracy. In panels A, B, and C, the accuracies shown are averages computed by comparison to the trainer's scores during five-fold cross validation. 888 worm tracks from both videos were divided into five equal groups for cross validation. (TIF)

**S7 Fig. Manual confirmation of lower nictation ratio in day 10 *C. elegans*.** All tracked *C. elegans* at 0, 15, and 30 min timepoints of all four replicates at each timepoint were examined and determined to be recumbent or nictating. Colored crosses represent the nictation ratio of 10–34 worms from one video, with each color representing one of the four biological replicates. Black crosses represent the average of the four video values at each timepoint. Vertical bars represent ±SEM. Day 10 NRs are significantly lower than the combined NRs from days 6 and 14 (p = 0.027, Wilcoxon Rank-Sum test). (TIF)

**S8 Fig. *C. elegans* centerline length over time.** The average length of the centerlines as calculated by the tracking code, excluding manually censored frames, is shown. Dots represent video averages, horizontal bars represent timepoint averages, vertical bars represent ± SEM. (TIF)

**S9 Fig. *C. elegans* nictation ratio decreases over the course of our 30 min videos.** The 10 s moving average of the nictation ratios of the four videos recorded at each timepoint is shown. The Pearson's correlation coefficient of the moving average NRs and video frame number is -0.0401, p = 0.00. (TIF)

**S10 Fig. Evaluation of Gaussian smoothing on *Steinernema carpocapsae* nictation videos.** **(A)** The effect of smoothing model output probabilities on the accuracy of common nictation metrics and overall transition rate. *Relative error* is the value calculated from the smoothed model probabilities divided by the value calculated from manual scores, minus one (equal to zero when there is no error). **(B)** The effect of smoothing model output probabilities on computer score accuracy on the validation data. **(C)** The effect of smoothing model output probabilities on the sum of the relative error of NR, IR, SR and 1—accuracy. In panels A, B, and C, the accuracies shown are averages computed from the validation set accuracies during five-fold cross validation. For *Steinernema*, 175 worm tracks from the same video were divided into five groups of 35 worm tracks for cross validation. σ = probability smoothing sigma, NR = nictation ratio, IR = initiation rate, SR = stopping rate, TR = transition rate, c. r. error = cumulative relative error. (TIF)

**S11 Fig. Choosing worm tracks for comparison. (A)** The number of worm tracks of at least 1 min duration being tracked at each frame in each video in the *S. carpocapsae* dataset. **(B)** The minimum number of worm tracks of at least 1 min duration being tracked in each video at each frame in the *S. carpocapsae* dataset. **(C)** The total number of worm tracks of at least 1 min duration being tracked in all the videos at each frame in the *S. carpocapsae* dataset. Among frames where at least two such worm tracks occur in every video, frame 3893 has the greatest total number of them, 106. (TIF)

**S1 Video. Tail wagging.** A *C. elegans* dauer from a liquid culture, 21 days post feeding, behaving on a microdirt arena. Frames 1–39 show normal nictation, followed by a transition period and then elevation and waving back and forth of the tail from frames 45–63. Afterward the

worm turns and begins normal crawling for the remainder of the clip. The video clip is cropped from the *C. elegans* testing video and was recorded at 5 fps. Pillars are in a 75 μm grid. (AVI)

## Acknowledgments

We would like to thank Anthony Fouad, Zihao Li, and Andre Brown for useful discussions concerning the design of the tracking and behavior classification algorithms; and Junho Lee, Heeseung Lee, Bram Cockx, and William Schafer for useful discussions regarding the project. The *C. elegans* strain N2 was provided by the CGC.

## Author Contributions

**Conceptualization:** Patrick D. McClanahan, Liesbet Temmerman.

**Data curation:** Patrick D. McClanahan.

**Formal analysis:** Patrick D. McClanahan.

**Funding acquisition:** Patrick D. McClanahan, Liesbet Temmerman.

**Investigation:** Patrick D. McClanahan, Luca Golinelli, Tuan Anh Le.

**Methodology:** Patrick D. McClanahan, Liesbet Temmerman.

**Project administration:** Patrick D. McClanahan, Liesbet Temmerman.

**Resources:** Liesbet Temmerman.

**Software:** Patrick D. McClanahan.

**Supervision:** Liesbet Temmerman.

**Validation:** Luca Golinelli, Tuan Anh Le.

**Visualization:** Patrick D. McClanahan.

**Writing – original draft:** Patrick D. McClanahan.

**Writing – review & editing:** Patrick D. McClanahan, Luca Golinelli, Tuan Anh Le, Liesbet Temmerman.

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
