## [Decision Letter · Decision Letter 0]

4 Jul 2023

PONE-D-23-13383Automated scoring of nematode nictation on a textured backgroundPLOS ONE

Dear Dr. McClanahan,

Thank you for submitting your manuscript to PLOS ONE. After careful consideration, we feel that it has merit but does not fully meet PLOS ONE’s publication criteria as it currently stands. Therefore, we invite you to submit a revised version of the manuscript that addresses the points raised during the review process.

We look forward to receiving your revised manuscript.

Kind regards,

Ebrahim Shokoohi

Academic Editor

PLOS ONE

Journal Requirements:

   "We would like to thank Anthony Fouad, Zihao Li, and Andre Brown for useful discussions concerning the design of the tracking and behavior classification algorithms; and Junho Lee, Heeseung Lee, Bram Cockx, and William Schafer for useful discussions regarding the project. N2 was provided by the CGC, which is funded by NIH Office of Research Infrastructure Programs (P40 OD010440)."

  "This work was supported by the Fonds Wetenschappelijk Onderzoek – Vlaanderen (https://www.fwo.be/, FWO G085521N awarded to L.T.) and KU Leuven (https://www.kuleuven.be, C16/19/003 awarded to L.T.). The funders had no role in study design, data collection and analysis, decision to publish, or preparation of the manuscript."

Additional Editor Comments:

Dear Authors

Please check the English style, and it must be significantly revised. The comments from the Referee also must be address. The revised version is attached for your reference. One of the author comment on the MS left before the submission.

Reviewers' comments:

Reviewer's Responses to Questions

**Comments to the Author**

1. Is the manuscript technically sound, and do the data support the conclusions?

Reviewer #1: Yes

2. Has the statistical analysis been performed appropriately and rigorously? 

Reviewer #1: Yes

3. Have the authors made all data underlying the findings in their manuscript fully available?

Reviewer #1: Yes

4. Is the manuscript presented in an intelligible fashion and written in standard English?

Reviewer #1: Yes

5. Review Comments to the Author

Reviewer #1: The manuscript contains very exciting information about nictation in free living nematodes. However, minor changes before acceptance needs to be addressed.

1-the English's style need to be revised. the comment's are attached.

2-What factors included in the behavior of nictation? What temperature, humidity, etc.?

3-What is the capability of in vivo application of this study?

4-Does it beneficial only to nematode behavior's?

5-How the nictation in the nature affect the behavior's of EPN and free living nematodes?

6. PLOS authors have the option to publish the peer review history of their article (what does this mean?). If published, this will include your full peer review and any attached files.

Reviewer #1: No

---

## [Author Response · Author response to Decision Letter 0]

12 Jul 2023

Response to editor comments:

We have reviewed the PLOS One formatting guidelines and set heading to the appropriate heading level as well as changed citations to the PLOS One style (reference numbers in brackets).

2. We note that you have provided funding information [in the Acknowledgements Section] that is not currently declared in your Funding Statement. However, funding information should not appear in the Acknowledgments section or other areas of your manuscript. We will only publish funding information present in the Funding Statement section of the online submission form. 

Please remove any funding-related text from the manuscript and let us know how you would like to update your Funding Statement. 

Thank you for pointing out this error. We have removed the funding source of the CGC from the Acknowledgments section of our manuscript (lines 724-728 with tracked changes visible) and requested that this information be included in the Funding Statement instead (please see cover letter).

Dear Authors

Please check the English style, and it must be significantly revised. The comments from the Referee also must be address. The revised version is attached for your reference. One of the author comment on the MS left before the submission.

We thank the editor for his comments. Our manuscript has been reviewed by a native speaker of English (see below).

Response to reviewer comments:

The manuscript contains very exciting information about nictation in free living nematodes. However, minor changes before acceptance needs to be addressed.

We thank the reviewer for their support and enthusiasm and for pointing out various ways we could improve our manuscript.

1-the English's style need to be revised. the comment's are attached. 

We thank the reviewer for their observations and corrections. Our revised manuscript has been checked by a native speaker of English. Based on their advice, we chose to keep these corrections in the revised manuscript, and also corrected several other grammatical and spelling errors. All these corrections can be seen as tracked changes in the 'revised_manuscript_with_changes_highlighted.docx' document appended to this submission.

2-What factors included in the behavior of nictation? What temperature, humidity, etc.? 

We agree that many factors may affect nictation behavior. This is why we performed all experiments in a climate-controlled room on microdirt arenas fabricated under controlled conditions. For S. carpocapsae, the temperature of the room was held between 23-24 °C and for C. elegans, 20-21 °C. For both, the humidity in the room was 30-45% (lines 578-579 of the revised manuscript). During all experiments, the arena was kept inside a sealed, humidified Petri dish (lines 586-588) and the same trans illumination system was used for recording (lines 554-559).

3-What is the capability of in vivo application of this study? 

We thank the reviewer for wondering about further potential applications of our system. Our study describes a behavioral assay performed on whole, living animals. As such, it could be used to assess the effects of a variety of interventions, in vivo in the nematodes. These include for example targeted mutations, transgene expression, gene knockdowns, compound interventions, varying environmental influences, etc. We have now mentioned such applications in the revised Discussion (lines 494-497).

4-Does it beneficial only to nematode behavior's? 

Our analysis pipeline is designed to score nictation in nematodes. However, it may be possible to apply the pipeline to other behaviors in other model organisms, provided they have a cylindrical shape whose posture can be represented as a spline (e.g. fruit fly larvae or zebrafish). This suggestion is included in the Discussion section of the revised manuscript (lines 504-509).

5-How the nictation in the nature affect the behavior's of EPN and free living nematodes? 

Nictation is a means for nematodes to attach themselves to larger animals, such as insects. EPNs and free-living nematodes differ in that for EPNs, the host is the food source, whereas for free-living nematodes, the host is a means of transport to a food source. We discuss this distinction in the manuscript Introduction (lines 35-40). How natural conditions such as CO2, air movement, and host cues affect this behavior is an important question, especially for biocontrol. Several studies have investigated the effects of various environmental conditions on nictation (cited in lines 389-391). We believe that our platform could be adapted to study these manipulations systematically in controlled conditions, and have added this to the revised discussion (lines 496-497).

We thank the editor and reviewer for their feedback. Additionally, we have corrected the y-tick labels on Fig 5C, which we noticed were wrong in our initial submission, and uploaded a new image files for that figure as well as the other figures after processing thru the PACE tool.

---

## [Editor Report · Decision Letter 1]

17 Jul 2023

Automated scoring of nematode nictation on a textured background

PONE-D-23-13383R1

Dear Dr.  McClanahan,

We’re pleased to inform you that your manuscript has been judged scientifically suitable for publication and will be formally accepted for publication once it meets all outstanding technical requirements.

Kind regards,

Ebrahim Shokoohi

Academic Editor

PLOS ONE

Additional Editor Comments (optional):

The authors addressed the questions raised by Referees and AE, as possible.

Reviewers' comments:

No comments

---

## [Editor Report · Acceptance letter]

24 Jul 2023

PONE-D-23-13383R1 

Automated scoring of nematode nictation on a textured background 

Dear Dr. McClanahan:

I'm pleased to inform you that your manuscript has been deemed suitable for publication in PLOS ONE. Congratulations! Your manuscript is now with our production department. 

Kind regards, 

on behalf of

Dr. Ebrahim Shokoohi 

Academic Editor

PLOS ONE